# On the Heterogeneous Distribution of Secondary Precipitates in Friction-Stir-Welded 2519 Aluminium Alloy

Ivan S. Zuiko [1], Sergey Malopheyev [1,*], Sergey Mironov [1], Sergey Betsofen [2] and Rustam Kaibyshev [1]

[1] Laboratory of Mechanical Properties of Nanoscale Materials and Superalloys, Belgorod National Research University, Pobeda 85, 308015 Belgorod, Russia; zuiko_ivan@bsu.edu.ru (I.S.Z.); mironov@bsu.edu.ru (S.M.); rustam_kaibyshev@bsu.edu.ru (R.K.)

[2] Moscow Aviation Institute, National Research University, Volokolamskoe Shosse, 4, 125993 Moscow, Russia; s.betsofen@gmail.com

[*] Correspondence: malofeev@bsu.edu.ru; Tel.: +7-4722-585-456

**Abstract:** The macro-scale distribution of secondary precipitates in friction-stir-welded 2519 aluminium alloy was studied. It was found that precipitation pattern essentially varied within the stir zone in terms of volume fraction, size, and even preferential concentration of the particles, either at grain boundaries or within the grain interior. This effect was attributed to local variations in welding temperature and cooling rate, which led to complex precipitation phenomena including coarsening, dissolution, and partial reprecipitation. Specifically, the precipitation coarsening was most pronounced at the weld root due to the lowest welding temperature being in this area. On the other hand, the highest welding temperature at the upper weld surface enhanced the dissolution process. The reprecipitation phenomenon was deduced to be most prominent in the weld nugget due to the slowest cooling rate being in this microstructural region.

**Keywords:** friction-stir welding; aluminium alloys; microstructure; second-phase precipitates

## 1. Introduction

Friction-stir welding (FSW) is an innovative joining technique that enables solid-state joining [1,2]. This technique is sometimes considered one of the most significant recent achievements in the field of joining, e.g., [3,4]. Nevertheless, one of the current challenges in this field is the welding of heat-treatable aluminium alloys. In this case, FSW typically leads to the dissolution of secondary precipitates in the stir zone [5–22], thus resulting in significant material softening [6,13,17,19,23,24].

To minimize this problem, several new FSW-derivative techniques have been invented recently, for instance, the stationary-shoulder FSW and bobbin-tool FSW. In the first approach, material mixing is conducted entirely by the tool probe, whereas the tool shoulder is kept stationary [25,26]. This gives rise to comparatively low heat input and provides a relatively uniform temperature distribution within the stir zone. In the bobbin-tool FSW, a specially designed, double-sided welding tool is used for joining [26]. The benefits of this technique include an essentially homogeneous temperature field within the weld zone, reduced axial force, and the elimination of the incomplete penetration defect.

Nonetheless, even the above-mentioned advanced techniques cannot overcome the softening problem in heat-treatable alloys completely. Hence, to recover alloys strength, the welded joints of these materials typically undergo a postweld heat treatment, which may involve solution annealing and subsequent artificial aging. The first step of this treatment, however, frequently promotes abnormal grain growth, i.e., the catastrophic coarsening of a few grains that eventually consume the entire stir zone [27–36].

It is important to emphasize that abnormal grain growth is frequently initiated from the stir zone periphery. Specifically, it preferentially develops either in the near-surface layer or at the weld root [27–29,32,37–42]. Although this effect is relatively well-known,

its origin is still not completely clear. It is sometimes believed that this phenomenon is associated with the specific microstructure (and, particularly, the specific precipitation pattern) that evolved in these areas during FSW. To the best of the authors' knowledge, however, this issue has not been studied experimentally so far. Attempting to shed some light on this topic, the present work is aimed at investigating the distribution of secondary precipitates within the stir zone of a typical heat-treatable aluminium alloy.

## 2. Materials and Methods

The material used in this work was a commercial 2519 aluminium alloy. It was produced by semicontinuous casting, homogenized at 510 °C for 24 h, swaged to a true strain of ≈2.0, and then rolled to a true strain of ≈1.4 at 425 °C. To produce the age-hardened condition (peak ageing), the rolled material was solution-annealed at 525 °C for 1 h, water-quenched, cold-rolled to a true strain of ≈0.2, and then artificially aged at 165 °C for 6 h. The obtained material was referred to as the base material.

The 3 mm thick sheets of the base material were butt-welded using a commercial AccurStir 1004 FSW machine. The welding tool was manufactured from tool steel and consisted of a concave-shaped shoulder of 12.5 mm in diameter and an M5 cylindrical probe of 2.7 mm in length. To evaluate the possible effect of the FSW heat input, two welding trials were conducted at different conditions, viz., (i) the low-heat-input condition and (ii) the high-heat-input one, as indicated in Table 1. These two regimes were selected on the basis of previous experiments. To evaluate the weld thermal cycle, K-type thermocouples were placed in close proximity to the stir zone border prior to FSW (Figure 1).

**Table 1.** FSW conditions applied in this work.

| Weld Designation | Tool Rotation Rate, rpm | Tool Travel Speed, mm/min |
|---|---|---|
| Low heat input | 500 | 760 |
| High heat input | 1100 | 380 |

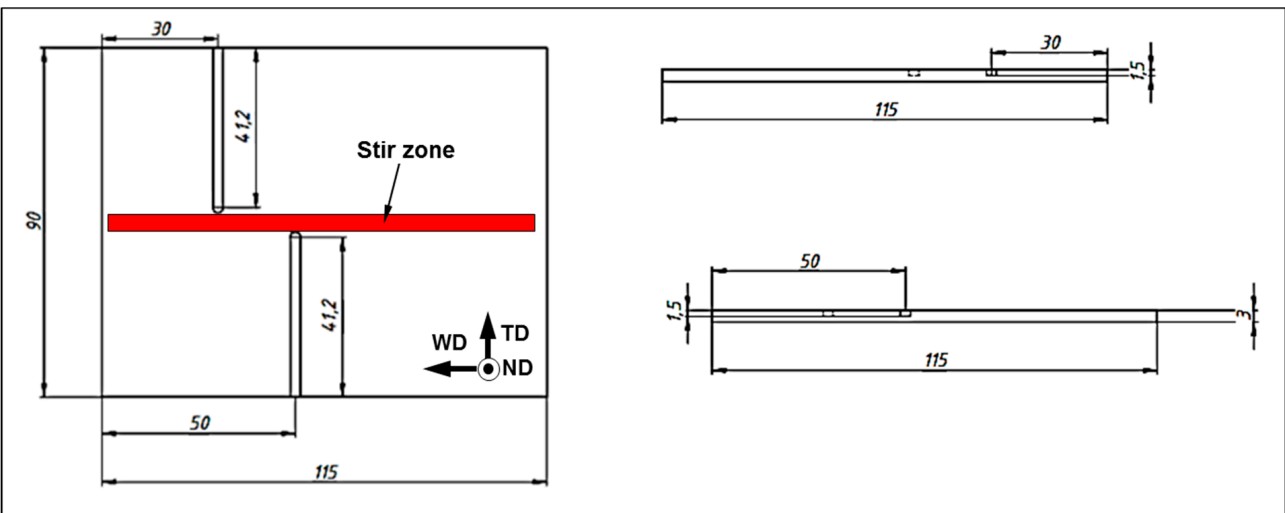

**Figure 1.** Thermocouple layout inside the workpiece (unit: mm). WD, ND and TD denote the welding direction, normal direction, and transverse direction. Not to scale.

Microstructural observations were focused on the examination of second-phase precipitates and were performed mainly by backscatter-electron-scanning electron microscopy (BSE-SEM). The microstructural specimens were prepared by mechanical polishing in conventional fashion, followed by vibratory polishing with fumed silica suspension OP-S (Struers, Copenhagen, Denmark). The BSE-SEM examinations were carried out using an

FEI Quanta 600 FEG-SEM, FEI Company, Hillsboro, OR, USA, operated at an accelerated voltage of 20 kV. To investigate the macro-scale distribution of the precipitates within the stir zone, three different microstructural regions were examined, viz., (i) upper section, (ii) weld nugget, and (iii) weld root (Figure 2 (In Figure 2, the significant lack of penetration was due to the insufficient plunge depth of the welding tool. In turn, this was associated with an attempt to provide the lowest temperature of welding as possible. This approach enabled important insight into the influence of the low-temperature FSW on microstructure)).

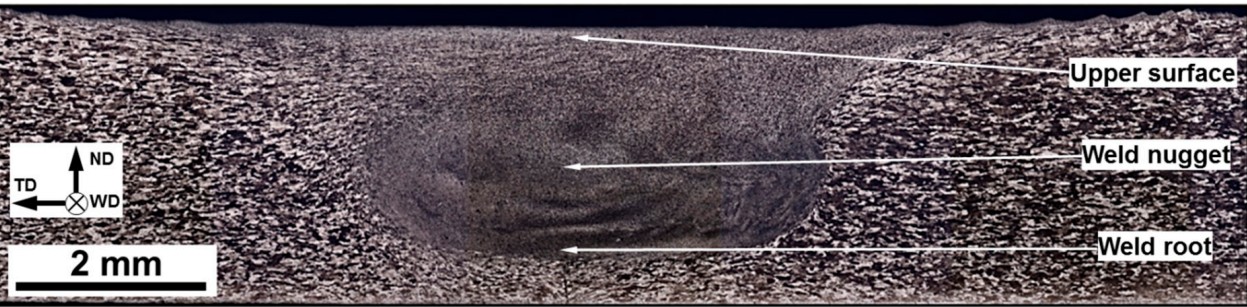

**Figure 2.** Optical image of the weld cross-section of the low-heat-input weld with indicated areas of microstructural observations. ND, TD, and WD are normal direction, transverse direction, and welding direction, respectively. In the micrograph, retreating side is left and advancing side is right.

The volume fraction and size distributions of the precipitates were quantified using image analysis software ImageJ (ver. 1.8.0, National Institutes of Health, Bethesda, MD, USA). Due to the penetration of the electron beam into the material depth, the acquired images also contained precipitates lying below the specimen surface. To exclude those from consideration, the micrographs were subjected to careful watershed segmentation in order to differentiate the precipitates from the matrix phase and the surface precipitates from the subsurface ones. The precipitation size was quantified using the equivalent-diameter method.

The BSE-SEM measurements were complimented by X-ray diffraction (XRD) examinations. These were conducted with the Rigaku SmartLab diffractometer, Rigaku Corporation, Tokyo, Japan equipped with a Cu K$\alpha$ radiation source in Bragg–Brentano geometry.

## 3. Results

### 3.1. Weld Thermal Cycle

As precipitation behaviour of heat-treatable alloys is a function of thermal conditions, microstructural analysis in the present study was preceded by the evaluation of the weld thermal history. The effect of FSW regimes on the measured thermal cycle is shown in Figure 3a. In good agreement with expectations, the high-heat-input weld exhibited a relatively high peak temperature and a comparatively long cooling time.

To facilitate interpretation of the precipitation behaviour, thermodynamic calculations were performed employing ThermoCalc 2020a software with the TCAl7 database (Figure 3b). As expected, FSW resulted in the dissolution of the constituent $Al_2Cu$ (i.e., $\theta$-) secondary phase, with this process being the most pronounced in the high-heat-input weld. Importantly, the dissolution was not completed in both welding conditions.

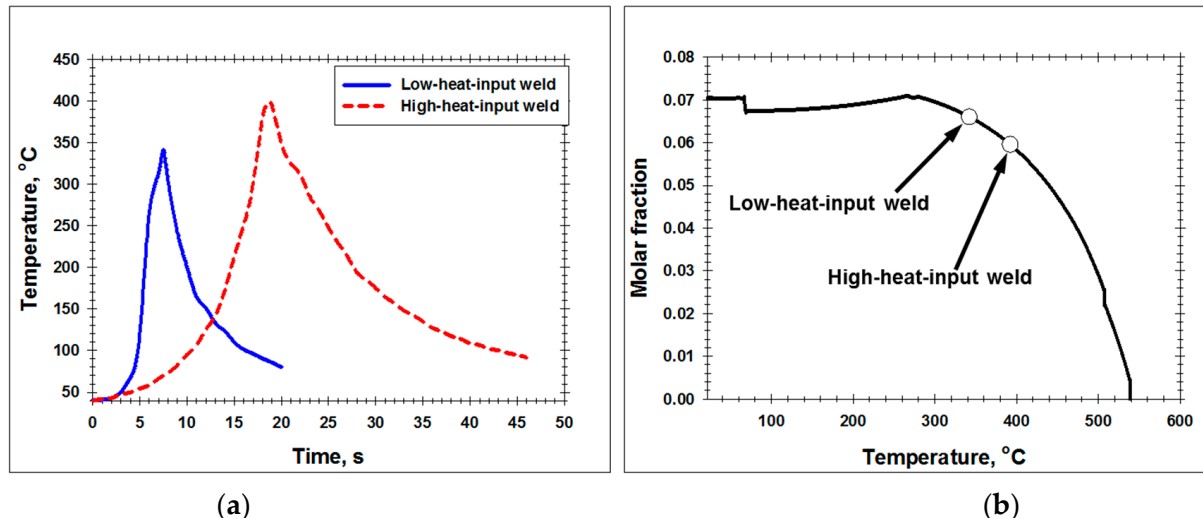

(a)                                                                      (b)

**Figure 3.** (**a**) Typical thermal cycles measured during FSW and (**b**) ThermoCalc prediction of the temperature dependence of the molar fraction of Al$_2$Cu (i.e., $\theta$-) phase.

### 3.2. Precipitation Pattern

The typical BSE-SEM images taken from different locations of the welded joints are shown in Figures 4 and 5. The corresponding precipitation statistics derived from the micrographs are summarized in Table 2 and Figure 6.

In the low-heat-input weld, the stir zone microstructure was dominated by the relatively coarse precipitates distributed more or less homogeneously on the scale of several grains (Figure 4). In the macro scale, however, a measurable difference in the volume fraction of the precipitates was found (Table 2). Specifically, the highest precipitation content was revealed in the weld nugget, whereas the lowest one was observed at the upper weld surface (Table 2). It is also worth noting that the particle-size distributions measured in these two local areas were characterized by a comparatively high percentage of fine-sized precipitates (Figure 6a).

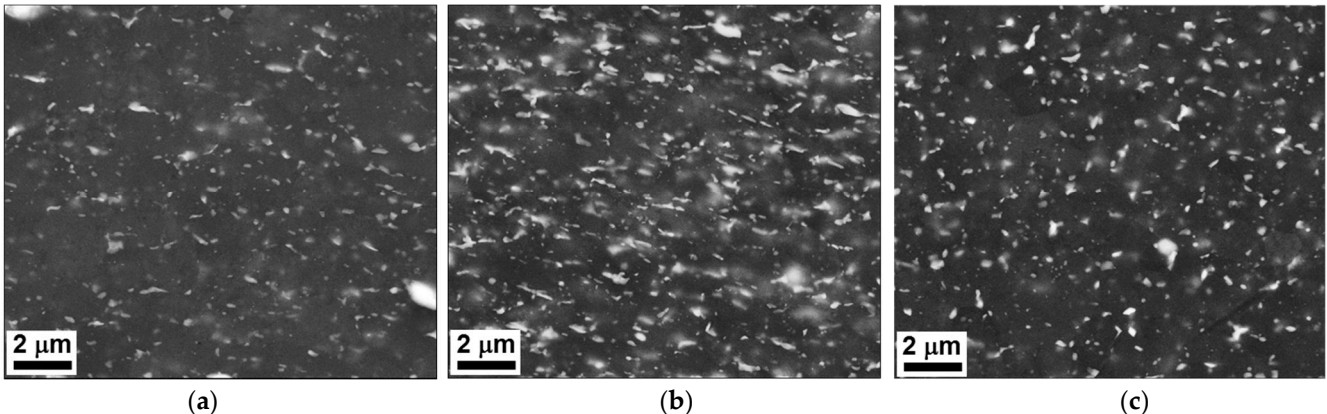

(a)                                         (b)                                         (c)

**Figure 4.** Typical backscatter-electron-scanning electron microscopy (BSE-SEM) images taken from different locations of low-heat-input weld: (**a**) upper surface, (**b**) weld nugget, and (**c**) weld root. Note: Secondary precipitates appear bright.

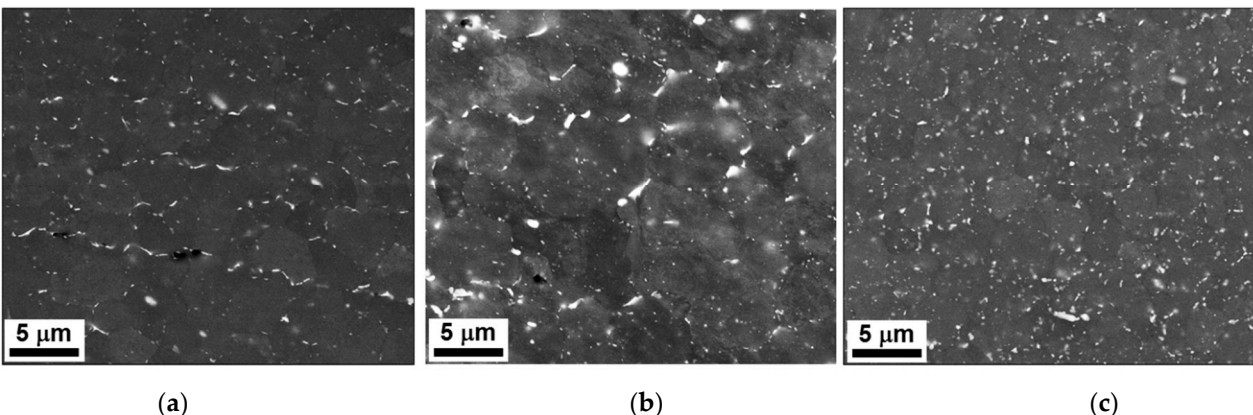

**Figure 5.** Typical backscatter-electron-scanning electron microscopy (BSE-SEM) images taken from different locations of high-heat-input weld: (**a**) upper surface, (**b**) weld nugget, and (**c**) weld root. Note: Secondary precipitates appear bright.

**Table 2.** Variation of volume fraction of secondary precipitates within stir zone (in vol.%).

| Particular Location within the Stir Zone | Low-Heat-Input Weld | High-Heat-Input Weld |
|:---:|:---:|:---:|
| Upper surface | 5.4 | 1.9 |
| Weld nugget | 6.7 | 2.3 |
| Weld root | 6.0 | 3.8 |

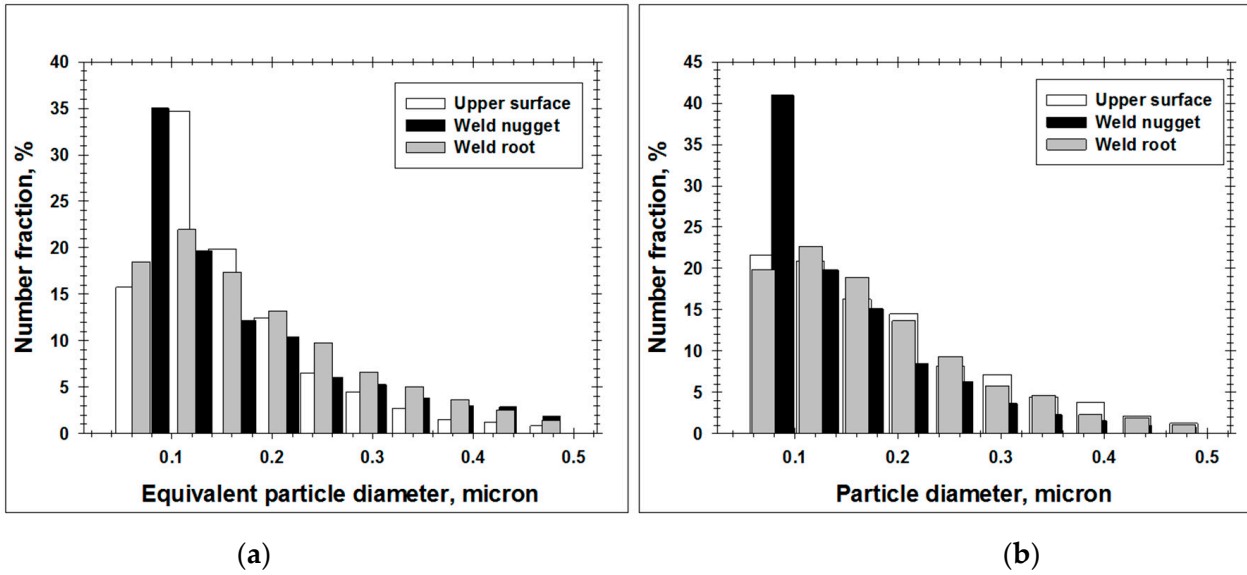

**Figure 6.** The precipitate-size distributions measured in different locations of (**a**) low-heat-input weld and (**b**) high-heat-input weld.

The high-heat-input weld exhibited an even more complex precipitation pattern. The material at the upper weld surface had the lowest precipitation content (Table 2). Importantly, the precipitates in this region were preferentially clustered along grain boundaries, being sometimes arranged as grain-boundary layers (Figure 5a). In the weld nugget, in addition to the grain-boundary precipitates, a significant fraction of the extremely fine dispersoids was also found in grain interiors (Figures 5b and 6b). As a result, this microstructural region exhibited somewhat increased precipitation content (Table 2). Finally, the weld root was characterized by the largest volume fraction of precipitates (Table 2), which were evenly distributed throughout the material (Figure 5c).

Significant grain refinement in both welding conditions was also noted. This effect is normally observed in friction-stir-welded/processed materials being most pronounced at the low-heat-input conditions, e.g., [43–47]. In heat-treatable aluminium alloys, however, grain refinement is typically insufficient to compensate for the FSW-induced precipitation dissolution, resulting in essential material softening in the stir zone, for example [43–47].

### 3.3. Phase Composition of Precipitates

In order to examine the typical phase composition of the precipitates, XRD measurements were applied (Figure 7). As expected, the precipitates were dominated by the Al$_2$Cu (or θ-) phase, which is the typical secondary phase in Al–Cu system. These results agreed well with our previous works [48–50]. Equally important, X-Ray data indicate the relatively low fraction of the θ-phase in the high-temperature weld, thus being consistent with BSE-SEM analysis (Table 2).

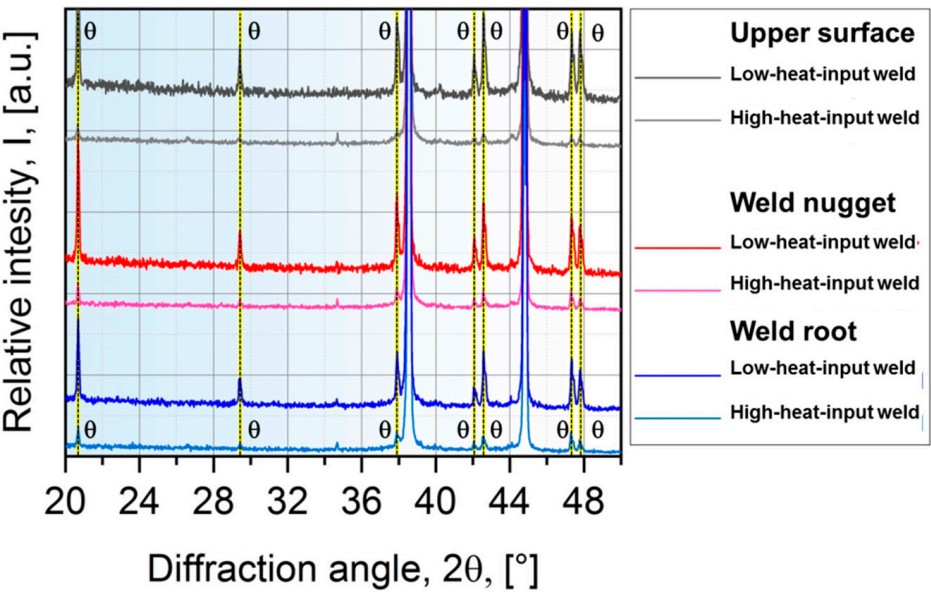

**Figure 7.** X-ray diffraction spectra taken from different locations of low-heat-input weld and high-heat-input weld. For clarity, the peaks associated with θ-phase are indicated by dotted lines.

## 4. Discussion

### 4.1. Uncertainties in the Temperature Measurements during FSW

Considering the substantial difference in precipitation patterns between the two studied welding conditions (Figures 4 and 5 and Table 2), it was surprising that the measured peak temperatures in these two cases were relatively close (Figure 3a,b). In this context, it is important to emphasize that the temperature measurements were conducted at the stir zone border (Figure 1), i.e., outside the stir zone. Given the significant thermal gradient inherent to FSW, it is highly likely that the actual temperature within the stir zone was essentially higher than that shown in Figure 3a. This raises some doubts regarding the relevance of the temperature measurements during FSW.

It is also noteworthy that the temperatures were measured only at the weld mid-thickness (Figure 1). Hence, the thermal history of the material at the upper weld surface as well as that of the weld root were virtually unknown. However, taking into account the well-accepted concept of the dominant role of the tool shoulder in the generation of FSW heat [1], it could be safely assumed that the highest welding temperature should be at the upper weld surface, whereas the lowest one should be at the weld root. Moreover, given the heat loss via air convention from the upper surface as well as the rapid heat sink into the backing steel plate, the highest cooling rate is expected at the weld root, whereas the lowest one is at the weld nugget.

*4.2. Precipitation Phenomena*

4.2.1. Low-Heat-Input Weld

The dominant microstructural characteristic of the low-heat-input weld was a considerable fraction of coarse precipitates within the stir zone (Figure 4). This observation implied a prevalence of precipitation coarsening rather than precipitation dissolution during low-temperature FSW. These findings are in good agreement with results recently reported by Kalinenko et al. [48].

On the other hand, considering the ThermoCalc predictions (Figure 3b) as well as the observed variation of the precipitation content across the stir zone (Table 2), some fraction of precipitates obviously went into the solid solution. As follows from the arguments given in Section 4.1, this process should be the least prominent at the weld root. Hence, this microstructural region should exhibit the highest precipitation content. In fact, however, the largest precipitation fraction was found in the weld nugget (Table 2).

The most plausible explanation for this observation is the partial reprecipitation of dissolved particles during the weld cooling cycle. Despite the relatively short duration of the cooling stage ($\approx$12 s in Figure 3a), the reprecipitation may perhaps occur because the average cooling rate ($\approx$21 °C/s) was significantly lower than that during water quenching. If so, the reprecipitation should be most prominent in the weld nugget due to the lowest cooling rate in this area.

The presumed reprecipitation of dissolved dispersoids agreed well with the increased fraction of the fine-sized precipitates revealed in the weld nugget and at the weld upper surface (Figure 6a).

4.2.2. High-Heat-Input Weld

In the high-heat-input weld, the increase in the welding temperature should promote precipitation dissolution. On the other hand, the prolongation of the cooling stage should enhance the reprecipitation process.

In this context, of particular interest was the grain-boundary nature of precipitates, which has been revealed at the upper weld surface (Figure 5a). Assuming that the grain structure in the stir zone was produced during FSW, the grain-boundary precipitates may only develop after FSW, during the weld cooling cycle, via reprecipitation from the solid solution. Due to the additional heat loss via air convection in this area, the cooling period near the upper weld surface was presumably relatively short. As a result, the reprecipitation occurred only in the local areas with enhanced diffusion activity, i.e., at grain boundaries.

A decrease in the cooling rate in the weld nugget should promote reprecipitation in the grain interior as well. Indeed, it has been observed in the present study (Figure 5b). The nano-scale nature of such precipitates (Figures 5b and 6b) was presumably associated with the limited duration of the reprecipitation process.

Considering the highest cooling rate in the weld root, the largest precipitation content in this area (Table 2) cannot be explained in the terms of the reprecipitation effect. Hence, it was likely attributable to the lowest welding temperature in this area, i.e., the incompletion of the precipitation dissolution process.

It is worth noting that precipitation coarsening and precipitation dissolution, both observed in the present study, were presumably essentially influenced by the severe plastic deformation occurring during FSW. However, the mutual contributions of thermal and strain-induced processes are unclear, and this issue requires further study.

**5. Conclusions**

This work was undertaken to investigate the macro-scale distribution of secondary precipitates in a typical friction-stir-welded heat-treatable aluminium alloy. To this end, 2519 aluminium alloy was used as a program material and two welding conditions were studied, viz., low-heat-input and high-heat-input. Microstructural observations were conducted with the BSE-SEM technique in three different locations within the stir zone:

(i) upper surface, (ii) weld nugget, and (iii) weld root. The main conclusions derived from this study were as follows.

(1) Temperature measurements were found to be not entirely consistent with microstructural observations. This discrepancy was attributed to the substantial temperature gradient inherent to FSW.

(2) The macro-scale distribution of the secondary precipitates within the stir zone was heterogeneous in terms of volume fraction, size, and even their preferential concentration at grain boundaries or within the grain interior. This effect was most pronounced under the high-heat-input welding conditions.

(3) The inhomogeneous character of the precipitate distribution was attributed to local variation in FSW temperature and cooling rate. These variations led to complex precipitation phenomena, including coarsening, dissolution, and even partial reprecipitation during the weld cooling cycle.

(4) The precipitation coarsening was found to be most pronounced at the weld root due to the lowest FSW temperature being in this area. On the other hand, the precipitation dissolution was found to be the most prominent at the upper weld surface because the highest welding temperature was in this region. The most intense particle reprecipitation was concluded to occur in the weld nugget due to the slowest cooling rate.

**Author Contributions:** Conceptualization, I.S.Z., S.M. (Sergey Malopheyev), S.M. (Sergey Mironov), S.B. and R.K.; methodology, I.S.Z. and S.M. (Sergey Malopheyev); validation, S.M. (Sergey Mironov), S.B. and R.K.; formal analysis, I.S.Z. and S.M. (Sergey Mironov); investigation, I.S.Z.; resources, S.M. (Sergey Malopheyev); data curation, S.M. (Sergey Mironov), S.B. and R.K.; writing—original draft preparation, I.S.Z. and S.M. (Sergey Mironov); writing—review and editing, S.M. (Sergey Mironov), S.B. and R.K.; visualization, I.S.Z.; supervision, S.B. and R.K.; project administration, R.K. All authors have read and agreed to the published version of the manuscript.

**Funding:** This research received no external funding.

**Institutional Review Board Statement:** Not applicable.

**Informed Consent Statement:** Not applicable.

**Acknowledgments:** The authors are grateful to the personnel of the Joint Research Center, "Technology and Materials" at Belgorod National Research University for experimental assistance.

**Conflicts of Interest:** The authors declare no conflict of interest.

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
