# Peer review of "On the Heterogeneous Distribution of Secondary Precipitates in Friction-Stir-Welded 2519 Aluminium Alloy"

_metals, doi:10.3390/met12040671_

Round 1

Reviewer 1 Report

Introduction to be focused more on FSW of precipitation strengthening Al alloys by addressing the challenges and research progress. So, you must refer some recent papers e.g. Weld zone and residual stress development in AA7050 stationary shoulder friction stir T-joint weld; Friction Stir Welding of Dissimilar Aluminum Alloy Combinations: State-of-the-Art; Stationary shoulder friction stir welding – low heat input joining technique: a review in comparison with conventional FSW and bobbin tool FSW. 

Weld cross section shows significant lack of penetration, which needs to explain further. It does not look like a good weld at all. I am wondering about investigating this weld cross section.

Suggesting to mention the grain structure correlatation with the tensile and hardness data in a comprehensive manner. Use some recent papers of grain refinement like: Mechanical and Damping Behavior of Age-Hardened and Non-age-hardened Al Alloys After Friction Stir Processing; Recent Development in Friction Stir Processing as a Solid-State Grain Refinement Technique: Microstructural Evolution and Property Enhancement; Through-thickness microstructure and mechanical properties in stationary shoulder friction stir processed AA7075; Investigation on the Effects of Welding Speed on Bobbin Tool Friction Stir Welding of 2219 Aluminum Alloy; Assessing the Bonding Interface Characteristics and Mechanical Properties of Bobbin Tool Friction Stir Welded Dissimilar Aluminum Alloy Joints.

Have you monitored the process temperature? Since it is precipitate strengthening heat treatable alloy, relying its properties on precipitation. 

Reviewer 2 Report

The work of this paper is of interest, but the manuscript needs some work for publication.

  1. Please give the deeply study on the influence of severe plastic deformation to secondary precipitates in friction-stir weld .
  2. The conclusion need complete revision. Some of the main results could be included in the conclusions.

Round 2

Reviewer 1 Report

Authors have revised the paper reasonably good, and now I recommend for publication.
